# Study on the Motion Characteristics of Solid Particles in Fine Flow Channels by Ultrasonic Cavitation

**DOI:** 10.3390/mi13081196

**Published:** 2022-07-28

**Authors:** Mu Yuan, Chen Li, Jiangqin Ge, Qingduo Xu, Zhian Li

**Affiliations:** College of Quality and Safety Engineering, China Jiliang University, Hangzhou 310018, China; yuanmu@zohomail.cn (M.Y.); gjq@cjlu.edu.cn (J.G.); kirkmoon@zohomail.cn (Q.X.); lizhian@cjlu.edu.cn (Z.L.)

**Keywords:** ultrasonic cavitation, fine particles, microfluidic, fluent simulation, bubble observation experiment, microfluidic mixing

## Abstract

Microjets caused by the cavitation effect in microchannels can affect the motion trajectory of solid particles in microchannels under ultrasonic conditions. The optimal parameters for an observation experiment were obtained through simulations, and an experiment was designed to verify these parameters. When the cavitation bubbles collapse in the near-wall area, the solid particles in the microchannel can be displaced along the expected motion trajectory. Using fluent software to simulate the bubble collapse process, it can be seen that, when an ultrasonic sound pressure acts on a bubble near the wall, the pressure causes the top of the bubble wall to sink inward and eventually penetrate the bottom of the bubble wall, forming a high-speed microjet. The maximum speed of the jet can reach nearly 28 m/s, and the liquid near the top of the bubble also moves at a high speed, driving the particles in the liquid towards the wall. A high-speed camera was used to observe the ultrasonic cavitation process of bubbles in the water to verify the simulation results. A comparison of particle motion with and without ultrasonic waves proved the feasibility of using the ultrasonic cavitation effect to guide small particles towards the wall. This method provides a novel experimental basis for changing the fluid layer state in the microchannel and improving precision machining.

## 1. Introduction

With the development of modern technology, the precision requirements of precision mechanical parts in various industries are continuously increasing. However, due to the small size and complexity of these precise workpieces, it is difficult for traditional precision machining methods to meet current size requirements. To overcome this technical issue, domestic and international experts have proposed the use of multiphase fluids mixed with solid abrasive grains and fluids for processing. Because of its “permeability” feature, this multiphase fluid has better accessibility and is therefore widely used in complex ultraprecision machining of fine structures. However, due to the small structure of some workpieces, the fluid moves in the fine internal flow channels as laminar flow, which makes it difficult to improve the surface accuracy of these workpieces after processing [1,2]. Fluid-based polishing methods adopt fluid to transport the particles, and the particle cutting motion is caused by the continuous movement of the fluid [3]. Therefore, studying the control method of multiphase fluids in fine flow channels is of great significance for improving the precision and efficiency of ultraprecision machining of tiny, complex workpieces.

The ultrasonic cavitation effect occurs when a large number of tiny bubble nuclei in a liquid undergo periodic expansion, compression, re-expansion, and recompression due to ultrasonic vibrations. After multiple oscillations, the large-volume cavitation bubble finally collapses at a high speed. Ultrasonic cavitation is a nonlinear process [4]. During liquid flow, the volume of the cavitation bubbles decreases rapidly as the surrounding pressure of the environment increases, and the collapse of the cavitation bubbles occurs in a few to hundreds of microseconds.

When a bubble is broken, many jets are ejected, or many daughter bubbles are formed [5]. Therefore, a large instantaneous pressure and flow velocity are produced in the area near the collapse [6]. The collapse of the cavitation bubble is generally accompanied by shock wave, microjet, hot spot, luminescence, and active radicals [7]. The collapse of the cavitation cloud in the fluid field produces a strong shock wave and a high-energy microjet near the microbubbles, which is expected to improve the degree of turbulence and the randomness of abrasive particle movement in the fluid field [8].

The wrinkle mechanism of a liquid sheet during a bubble collapse has attracted skyrocketing attention from various fields, including mathematics and material science [9,10,11]. According to the theory of cavitation collapse and the characteristics of the fluid near the wall, near the rigid wall, the pressure on the opposite side of the wall is relatively large, which causes the bubbles to dent and eventually form a microjet that impacts the wall. Further research has revealed that, when the cavitation bubbles near the wall collapse, the bubble wall as a whole shrinks inward. Ultrasonic vibrations in the water effectively disrupt fluid motion and increase the kinetic energy of the particles in the water [12,13,14]. Acoustic streaming increases the local fluid dynamic pressure, and the collapse of the cavitation bubbles produces numerous microjets, which increases the kinetic energy of the fluid near the wall [15]. At this time, liquid flows towards the center of the bubble, but, because the liquid near the wall is blocked by the wall, the speed is lower than the flow speed of the liquid on the opposite side of the wall, which eventually causes the bubble wall on the opposite side of the wall to sink. Eventually, a micro jet that impacts the wall surface is generated.

At present, the ultrasonic cavitation effect has been applied to precision machining fields such as EDM (Electrical Discharge Machining) and material deburring. Hloch [16], et al. found that the utilization of ultrasonic pulsating jet can accomplish the fracture-free removal of bone cement. Liew et al. [17] used ultrasonic cavitation-assisted EDM to machine microholes and found that the oscillation of the cavitation bubbles can significantly increase the discharge speed of debris and reduce the adhesion of debris on the workpiece surface. Timing Ming et al. [18] used the ultrasonic cavitation effect to strengthen the impact of abrasive flow on the surface of the workpiece to be processed and reduced the surface roughness of the workpiece. Toh et al. [19] studied the effect of ultrasonic cavitation on burr formation and burr height during stainless steel milling and found that ultrasonic cavitation can effectively reduce burr height. Ye Linzheng et al. [20] analyzed the impact characteristics of the honing cavitation bubble collapse microjet with the SPH-FEM coupling method and used orthogonal experiments to discover that the surface quality of the cavitation honing workpiece improved under the appropriate parameters. Zhu et al. built the particle motion model in the gas purging process by coupling volume of fluid (VOF) method with the discrete phase model, and calculated the oil–gasflow distribution and the wall erosion rate [21]. Many studies have shown that ultrasonic cavitation-assisted processing has promising characteristics. On this basis, the microjet produced by cavitation was used to act on the solid particles in the flow channel to affect their motion trajectory, changing the fluid layer state, improving precision machining accuracy, and providing new solutions for various other problems.

This paper takes the multiphase flow field in microflow channels as the research object and studies the motion state of solid particles in the flow field after cavitation. In this paper, the Navier–Stokes equation was combined with the VOF model to perform finite element numerical simulations of gas-liquid two-phase motion, and the PISO algorithm was used to solve the problem. The cavitation bubble collapse process in the microchannel was simulated and studied to determine the optimal parameter values of the initial cavitation bubble, such as the radius and the wall distance to the bubble. Finally, an ultrasonic cavitation observation experimental platform that simulated the fine flow field was built, and the cavitation bubble collapse evolution process and the effect of the cavitation bubbles on the movement of the solid particles were observed and tested. The observation results show that the use of microjets generated by the collapse of cavitation bubbles near the wall affects the solid particles in the flow channel, effectively driving the nearby particles to produce directional motion and stimulating the flow field to break the laminar flow pattern and improve the motion of the fine particles. The increased probability of hitting the wall effectively improves the precision and efficiency of ultraprecision machining.

## 2. Numerical Theoretical Analysis of Cavitation Bubbles

### 2.1. Cavitation Bubble Dynamics Model

The cavitation effect is a series of processes that occur during the formation, expansion, contraction, and collapse of vapor or gas bubbles in the liquid. After the collapse, the cavitation bubbles produce microjets that move throughout the liquid. Therefore, the microfluidic jet generated in the microchannel can effectively affect the movement trajectory of the solid particles in the channel. If the cavitation effect occurs near the wall, the generated microjet more effectively causes the fine solid particles to impact the wall surface, thereby improving the precision and efficiency of ultraprecision machining. The cavitation phenomenon of the bubbles in the liquid can be regarded as the movement process of the bubble wall [22]. Assuming that the liquid is incompressible, the temperature of the liquid is constant, and the intensity of the sound field remains constant; the microbubbles (considered spherical symmetrical motion) in the liquid will be affected by the tensile and compression phases of the ultrasonic waves. Considering the effect of the liquid viscosity and surface tension on bubble motion, the motion equation of the cavitation bubble wall under the action of multiple parameters can be derived from the conservation of energy equation [23,24,25,26]:(1)$$R\frac{d^{2}R}{dt^{2}}+\frac{3}{2}{\left(\frac{dR^{2}}{dt}\right)}^{2}=\frac{1}{\rho }\left[\left(P_{0}+\frac{2\sigma }{R_{0}}-P_{\nu }\right)\right.{\left(\frac{R_{0}}{R}\right)}^{3k}+\left.P_{\nu }-P_{0}+P_{A}\sin2\pi ft-\frac{2\sigma }{R}-\frac{4\mu }{R}\frac{dR}{dt}\right]$$
where *R* is instantaneous bubble radius; *R*_0_ is initial bubble radius; *P_v_* is vapor pressure in bubble; *P_A_* is sound pressure amplitude; *P*_0_ is standard atmospheric pressure; *σ* is surface tension; *μ* is viscosity of the reaction system; *k* is adiabatic index; *f* is ultrasonic frequency; *ρ* is liquid density; and *t* is cavitation bubble movement time.

The stimulating factors of the cavitation effect include high-intensity ultrasonic radiation, violent vibrations during explosions, the impact friction of high-speed fluids, and violent chemical reactions. Therefore, the dynamic model of ultrasonic excitation used in this paper does not include radiation damping.

The gas–liquid–solid three-phase abrasive flow is a typical multi-scale physical system, in which the macro scale of the continuous fluid phase and the micro scale of bubble collapse coexist. In the three-phase abrasive flowfield, the energy generated by bubble injection and collapse is the key to improve the polishing efficiency [27]. The fluid turbulent motion also can be considered as the process of generation, development, and dissipation of multi-scale turbulent vortices [28]. As previously mentioned, the ultrasonic cavitation effect is a series of nonlinear processes that occur during the expansion and collapse of a gas in a liquid. The movement of the bubble wall in the fluid is the focus of this article. The multiphase flow problem can be solved by the VOF model in the multiphase flow model.

The Euler model is a more complicated model. Each phase in the control volume is solved by establishing a continuum equation and momentum equation with *N* parameters. The exchange coefficient between the interfaces is coupled to the pressure term according to the type of phase contained and the internal conditions of the fluid [29].

The VOF model uses Euler’s method and includes a continuum model of multiple fluids. Two or more fluids share a set of momentum equations, and the volume fraction of each fluid in the calculation domain is tracked for each calculation unit. Some applications of this model include stratified flow, bubble flows in liquids, and the steady-state or transient interface liquid–gas problems in this paper.

At the same time, to study the evolution process of the bubble wall in the flow channel under ultrasonic action, the Navier–Stokes equation was combined with the VOF model to simulate gas–liquid two-phase motion, and the momentum equation and continuity equation are:(2)$$\frac{\partial \rho u}{\partial t}+\left(u\cdot \nabla \right)\rho u=-\nabla \rho +\mu \nabla^{2}u$$
(3)$$\frac{\partial \rho }{\partial t}+\nabla \cdot \left(\rho u\right)=0$$
where *u* is fluid velocity vector; *ρ* is fluid density; *μ* is fluid viscosity.

### 2.2. Analysis of Influencing Factors of Microjets Generated by Cavitation Bubble Collapse

The three main influencing factors of microfluidic jets generated by cavitation bubble collapse are a dimensionless quantity, the cavitation bubble radius and the ultrasonic sound field.

#### 2.2.1. Influence of the Dimensionless Quantity on Cavitation Microfluidic Jets

To study the influence of the distance between the bubble and the wall on the cavitation effect, a dimensionless quantity *γ* was defined to indicate the distance from the bubble to the wall. The dimensionless quantity can be expressed as:(4)$$\gamma =\frac{L}{R}$$
where *L* is the distance from the center of the bubble to the wall. *R* is the initial radius of the bubble.

It is known through research that, when the other conditions are held constant, the larger *γ* is, the greater the distance between the cavitation bubble and the wall surface. At the same time, the smaller the dimensionless distance *γ* is, the greater the impact velocity and impact effect of the cavitation bubble collapse jet on the wall, and the greater the probability that the particles driven by the jet will break. Therefore, in this paper, while the other influencing parameters remain constant, appropriate dimensionless quantities should be selected to simulate the collapse process.

#### 2.2.2. Effect of the Cavitation Bubble Radius on Cavitation Microfluidic Jets

The bubble radius is a very important factor for cavitation. The inside of the bubble has vapor pressure and gas pressure, while the outside has ultrasonic pressure, hydrostatic pressure and surface tension. By appropriately simplifying the motion process of the bubble wall, the Rayleigh–Plesset equation of motion can be obtained:(5)$$R\frac{d^{2}R}{dt^{2}}+\frac{3}{2}{\left(\frac{dR}{dt}\right)}^{2}=\frac{1}{\rho }\left(P_{B}-P_{0}+\frac{2\sigma }{R_{0}}-\frac{4\mu }{R}\frac{\mathrm{dR}}{dt}\right)$$
where *P*_A_ is initial bubble radius; *P*_A_ is ultrasonic pressure; *P*_0_ is hydrostatic pressure; *μ* is viscosity coefficient; *k* is polytropic index; *R* is actual bubble radius; and *σ* is surface tension coefficient.

When the bubble radius is 5 μm or 500 μm, the bubble wall will periodically oscillate under the action of ultrasound. When the bubble radius is 50 μm, the bubble can be cavitated in less than one ultrasound cycle. If the bubble radius is too large or too small, it does not collapse easily [30]. To study the influence of different bubble radii on the collapse of the bubble, this paper selects appropriate sizes for the bubble radius to simulate the collapse process while leaving the other influencing parameters unchanged.

#### 2.2.3. The Effect of an Ultrasonic Sound Field on Cavitation Microfluidic Jets

Different ultrasonic sound fields will lead to different degrees of difficulty or intensity of cavitation phenomenon. Therefore, the ideal test results can only be achieved by selecting appropriate sound field parameters according to different experimental conditions. The sound field parameters are mainly ultrasonic frequency *f* and sound intensity I.

When the intensity of sound field is constant, the higher the ultrasonic frequency is, the more difficult cavitation occurs. As the ultrasonic frequency increases, the duration of ultrasonic stretching phase will be shortened, and the effect on cavitation core is not enough. The cavitation core cannot grow to a sufficient size and obtain sufficient energy, so it cannot collapse. Therefore, the increase of ultrasonic frequency will weaken the cavitation effect. In addition, high-frequency ultrasound loses energy faster as it travels through liquids, so it takes more energy to get the same chemical effect.

At the same time, the cavitation effect can be strengthened by increasing the sound intensity of ultrasonic wave. When the selected ultrasonic frequency can not produce cavitation bubbles, the cavitation bubbles can be excited by increasing the sound intensity. In addition, sound intensity also affects the collapse time, the maximum pressure Pmax, and the maximum temperature Tmax generated by the collapse.

Through the above analysis, it can be seen that ultrasonic frequency and sound strength are the main factors affecting cavitation. In the experimental study, ultrasonic frequency should not be too high, usually 20–40 KHz. The sound must be strong enough to produce cavitation.

## 3. Ultrasonic Cavitation Model and Numerical Simulation

### 3.1. Model Building

In this paper, Ansys Fluent 19.0 software (Ansys Fluent, version 19.0, accessed on 5 May 2022) was used to numerically simulate the cavitation effect of bubbles in the area near the wall of the micro flow channel under the action of ultrasonic waves. Since the cross-sectional size of the micro flow channel is at the micron level, a 500 μm × 500 μm square water area simulation was established. An air bubble was created in the bottom area of the cross-section of the micro flow channel, and the bottom surface was set as a rigid wall surface. Because the liquid flows slowly in the microfluidic chip and the time for the ultrasonic cavitation effect to occur is extremely short, the liquid is considered to be in a static state. The initial velocity of the water in the control body was set to 0, and the upper side was set as the pressure inlet.

As shown in Figure 1, in the established physical model, the initial bubble radius *R*_0_ and ultrasonic frequency f have the greatest impact on the calculation results, so different combinations of these working conditions were used for multiple calculations.

Through the reading and reference of a large number of literature and the database data in the simulation software, some hypothesis data which have guiding significance for the important parameters in the follow-up observation experiment are drawn up [2,9]. The specific calculation conditions are shown in Table 1.

### 3.2. Platform Basic Meshing and Calculation Setting

The geometric model in this article uses Gambit software (Gambit, version 2.4, accessed on 20 May 2022) for meshing. To accurately obtain the movement of the bubble wall while reducing the overall calculation amount, when the physical model is meshed, the edge area of the geometric model needs to be sparsely divided. The dotted line indicates the location of the bubble, and the area near the bubble is encrypted.

As shown in Figure 2, the minimum grid was 1.312605 × 10^−13^ m^2^, and the number of grids was 57,845, For a square fluid field with a cross section of 500 μm, the model simulation contains bubbles with a radius of 50 μm, which meets the requirements of the simulation test. The model used the pressure-based transient solution method, and the PISO (pressure-implicit with splitting of operators) algorithm was used to couple pressure and velocity. In the discrete format, the least squares format was used for the gradient item; the Pressure Staggering Option format was selected for the pressure item; the momentum item, density item, and energy item were all selected in the second-order upwind style; and the modified Geo-Reconstruct format was used for the volume fraction item. The time term used the first-order implicit format. The momentum equation and the continuity equation used 10^−6^ as the convergence condition, the energy equation used 10^−7^ as the convergence condition, the calculation time step was selected as 5 × 10^−7^ s, and each iteration was set to 40 times.

### 3.3. Analysis of Results

This simulation was based on the Ansys Fluent platform. The simulation used the VOF model to simulate and calculate the collapse process characteristics of bubbles near the wall in an ultrasonic sound field and investigated the initial bubble radius *R*_0_, the ultrasonic frequency f, and the maximum velocity of the microjet generated by the bubble collapse. From the calculation results of various working conditions under different parameter combinations, it was determined that, for an initial bubble radius *R* = 100 μm, *γ* = 1.6, a sound pressure intensity of 300 kPa, and an ultrasonic frequency *f* = 20 kHz, the generated microjet velocity was the largest. Thus, the working conditions under this parameter combination were selected as an example to analyze the characteristics of the bubble collapse process and conduct experimental observations.

As shown in Figure 3a–f, the process of the bubble wall from the beginning of the depression to its complete collapse was recorded. In each figure, the left image shows the volume fraction cloud image, and the right image shows the velocity vector diagram. As shown in Figure 3a, at *t* = 32.5 μs, the pressure of the ultrasonic sound field acts on the upper part of the bubble wall, resulting in a higher pressure outside the upper part of the bubble, and the upper part of the bubble begins to shrink inward as a whole. As shown in Figure 3b, at *t* = 37.5 μs, as the pressure on the outside increases, the downward movement speed of the top of the bubble wall is much higher than that of other parts, and the top of the bubble rapidly collapses downward. As shown in Figure 3c, at *t* = 42.5 μs, the bubbles appear inwardly concave, forming micro-jets, and the maximum velocity is close to 28 m/s. As shown in Figure 3d, at *t* = 47.5 μs, the pressure inside the bubble gradually increases as the volume shrinks, and the rate at which the top of the bubble wall collapses down also begins to decrease. As shown in Figure 3e, at *t* = 52.5 μs, the high-speed micro-flow of the bubble jet to the wall begins to affect the flow of liquid between the bubble and the wall. The liquid on the bottom surface that originally flowed to the bubble wall became short-lived and almost static. As shown in Figure 3f, at *t* = 57.5 μs, the microjet caused by the collapse of the bubble wall pushes the liquid at the bottom to move away from the bubble wall. At this time, the top of the bubble wall completely penetrates the bottom, the bubble divides in two, and a microvortex forms between the two small bubbles.

At the same time, during the collapse, the downwards movement of the bubble wall causes the water to move away from the surface and downwards with the bubble, and the entire bubble is pushed closer to the wall. The top of the bubble is the first part to be affected by the sound pressure, which moves rapidly towards the wall, and the position of the maximum speed approaches the wall. At the same time, during the collapse, the downwards movement of the bubble wall causes the water to move away from the surface and downwards with the bubble, and the entire bubble is pushed closer to the wall. The top of the bubble is the first part to be affected by the sound pressure, which moves rapidly towards the wall, and the position of the maximum speed approaches the wall.

To study the effect of the ultrasonic frequency on the maximum velocity of the cavitation microjet, a frequency of 20 kHz was selected while the initial bubble radius *R*_0_ = 50 μm, *γ* = 1.4, and ultrasonic sound pressure *P* = 300 kPa remained constant during the simulation. Three frequencies (20, 30, and 40 kHz) were simulated three times each to determine the relationship between the ultrasonic frequency and the maximum velocity of the microjet when the bubble collapsed.

At the same time, to study the influence of the bubble radius on the maximum velocity of the cavitation microjet, 50 μm was selected while the ultrasonic frequency *f* = 20 kHz, *γ* = 1.4, and the ultrasonic sound pressure *P* = 300 kPa remained constant during the simulation. Three radii (50, 75, and 100 μm) were used as the initial radius of the bubble to determine the relationship between the initial radius of the bubble and the maximum velocity when the bubble collapses.

As shown in Figure 4, as the bubble radius increases from 50 μm to 100 μm, the maximum velocity of bubble collapse gradually increases, but the difference between the maximum velocity values is not large. This is because, under the action of ultrasonic sound pressure, the larger the bubble radius is, the greater the initial potential energy of the bubble, and the faster the bubble shrinks when it releases the potential energy; thus, the velocity of the microjet increases. Therefore, in the experiment, the initial radius of the bubble was selected as 100 μm to achieve better experimental results.

As shown in Figure 5, as the ultrasonic frequency increases, the maximum velocity of bubble collapse gradually decreases. Ultrasonic cavitation is a process in which bubbles accumulate potential energy during the tensile phase of ultrasonic waves and release potential energy during the compression phase. As the frequency of the ultrasonic wave increases, the duration of the stretching phase decreases, and the bubbles cannot accumulate a large amount of potential energy; thus, when the potential energy is released, the velocity of the generated microjet decreases.

## 4. Ultrasonic Cavitation Bubble Observation Test

### 4.1. Ultrasonic Cavitation Observation Test Platform

Figure 6 shows the experimental platform for the observation of the ultrasonic cavitation bubble collapse. The models of the experimental instruments are:Ultrasonic transmitter: produced by Shanghai Qixun Instrument Co., Ltd. (Shanghai, China), model QUN-650Y. The rated power is 650 W (the actual power is adjustable from 1% to 99%). The equipped horn has a diameter of 6 mm, a vibration frequency of 20 kHz, and an error of ±1.0 kHz.Microfluid syringe pump: produced by Harvard Company, (Boston, MA, USA); the model is Pump 11 Elite. This syringe pump can be used with a 0.5 μL to 50 mL syringe; the delivery speed range is 1.28 pl/min to 88.28 mL/min, and the delivery accuracy is ±0.5%.High-speed camera: Produced by Japan NAC Company (Tokyo, Japan); the model is MEMRECAM HX-6. NAC MEMRECAM HX-6 is a high-resolution, high-speed, high-sensitivity camera. It has a resolution of 2560 × 1920 pixels and a maximum shooting speed of 210,000 fps. Because the camera’s memory is limited, a single shot can only record thousands of photos. Therefore, in high-frequency shooting, the recording time is very short, so it is usually necessary to use a delay trigger.Light source: produced by Japan Altec system company (Tokyo, Japan); the model is LLBK1-LA-W-0022, with a power of 160 W. There is a high-concentrating convex lens in front of the light source that focuses the light emitted by the light source to the focal point and improves the local brightness. This is necessary for experiments with poor reflectivity, such as shooting bubbles in water.Superabsorbent polymer particles: the superabsorbent polymer particles were in a translucent state, contained strong hydrophilic groups, and can absorb thousands of times their own weight of water. These particles swell after absorbing water and have good elasticity.

Test process: Set the parameters to the optimal parameter values obtained through the simulations: an ultrasonic sound field frequency of 20 kHz, an initial cavitation bubble radius of 100 μm, and a dimensionless quantity *γ* of 1.6. Then, turn on the light source and the high-speed camera. Aim the camera at the sink and take a quick shot after launching the ultrasonic waves and generating the cavitation. To observe the test results more clearly and conveniently, a high-speed camera with a shooting frame rate of 50,000 fps was selected; the microsyringe pump was allowed to push the syringe at a speed of 10 μL/min, and the syringe pump was paused when the radius of the bubble generated by the needle tip was approximately 100 μm. Let the bubble adhere to the needle tip; slowly move the needle tip closer to the wall and stop moving the tip when the distance from the bubble wall to the wall surface is less than approximately 1/3 the bubble diameter.

### 4.2. Ultrasonic Cavitation Observation Experiment and Result Analysis

Based on ultrasonic cavitation bubble collapse theory, three sets of observational comparison experiments were carried out to verify the bubble collapse under the action of ultrasonic waves near the wall, investigate the effect of the bubble collapse on the surrounding particles, and observe the influence of the bubbles on the particles moving in the flow channel as ultrasonic cavitation collapse occurs. Figure 7 shows the observation experiments to verify the cavitation collapse of bubbles near the wall under the action of ultrasonic waves. The bubble is pushed out of the needle tip by the ultrasonic wave; this time was taken as 0. The bubble moves and touches the wall at *t* = 40 μs; due to inertia, the bubble is further compressed in the lateral direction and elongated in the longitudinal direction, resulting in an elliptical shape. At *t* = 80 μs, the bubble rebounds, and under the action of the ultrasonic stretching phase, the bubble widens in the lateral direction, and the left side of the bubble wall becomes “pointed”. At *t* = 100 μs, the ultrasonic compression phase affects the bubble wall. The left side of the bubble wall quickly dents towards the wall surface and collapses, forming a microjet that impacts the wall surface. The figure shows that, from *t* = 80 μs to *t* = 100 μs, the displacement of the bubble wall collapse is approximately 300 μm, and the time taken is 20 μs. The calculation shows that the bubble collapse speed is 15 m/s, which is similar to the simulation results. From these experiments, it can be found that the collapse process of bubbles under the action of ultrasonic waves is similar to the simulation results.

Figure 8 shows observation experiments to explore the influence of the bubbles on the surrounding particles when the cavitation collapses under the action of ultrasonic waves. Superabsorbent polymer particles with good elasticity were added to the test to observe the effect of the bubble collapse on the surrounding particles. The main component of super-polymerization water-absorbing particles is polyacrylate sodium salt. It is used to simulate solid particles in flow field in precision machining. To facilitate this observation, the superabsorbent polymer particles were hooked by fine silver wires and placed near the bubbles to prevent the particles from sinking. As shown in the figure, the superabsorbent polymer particles were placed on the side of the bubble away from the wall and were approximately 100 μm away from the bubble. Due to the constraint of the silver wire, the particle moves to the wall more slowly than the bubble, and the distance between the particle and the bubble increases. At *t* = 0 μs, the bubble reaches the wall and continues to compress in the horizontal direction due to inertia. At *t* = 20 μs, the bubble stretches laterally due to rebound and ultrasonic action; at the same time, the particles continue to be pushed by the ultrasonic wave. When the bubble moves at a lower speed, the presence of the particles does not significantly affect the cavitation process of the bubble. At *t* = 60 μs, the next compressed phase of the ultrasonic wave reaches the bubble wall and starts to move the bubble wall towards the wall. The side of the bubble opposite the wall surface quickly sinks towards the wall surface until it collapses. At *t* = 80 μs, it can be clearly observed that the rapid movement of the bubble wall forms a large pulling force on the side of the bubble wall opposite the rigid wall surface, and superabsorbent polymer particle deformation occurs under the action of the pulling force. In the experiment, due to the restraint of the silver wire microhooks, the particles did not move to the wall but instead deformed. However, it can be inferred that, if there was no silver wire constraint, the particles would move towards the wall as the bubble collapses.

Figure 9 shows the relative distances between the cavitation bubbles and the solid particles in the microchannel without ultrasonic waves. According to the data, when the ultrasonic waves are not applied, the liquid in the microchannel is a smooth laminar flow, the cavitation bubbles and the solid particles are relatively static, and the bubbles cannot cause cavitation collapse.

Figure 10 shows the relative distance between the cavitation bubbles and the solid particles near the wall of the microchannel under the action of ultrasonic waves. According to the data, after applying the ultrasonic waves, the bubbles touch the wall due to the water flow and the ultrasonic sound field. Due to inertia, the bubbles continue to compress and become ellipsoidal, and the particles gradually approach the bubbles. At *t* = 100 μs, the bubble begins to rebound, accelerates under the influence of the ultrasonic stretching phase, and elongates in the longitudinal direction. After that, the compression phase of the ultrasonic wave pushes the upper part of the bubble wall downwards, collapsing it and forming a high-speed microjet, which disrupts the laminar flow of the liquid around the cavitation bubble and drives the nearby particles to move quickly towards the wall surface. From the observation results, it can be seen that the micro-jet formed by the collapse of cavitation bubbles can affect the motion characteristics of solid particles in the surrounding flow field. At the same time, the cavitation bubble collapsing on the wall can compress the fluid field and guide the surrounding solid particles to move toward the wall to a certain extent, so as to improve the probability of solid particles hitting the wall.

When Figure 9 and Figure 10 are compared, it can be seen that the absence of the ultrasonic wave and the application of an ultrasonic wave in the vicinity of the wall near the microchannel have a considerable impact on the motion trajectories of the solid particles and cavitation bubbles. We extracted and compared the relative displacement between the particles and the cavitation bubbles from 0 to 100 μs, as shown in Figure 11.

As shown in Figure 11, the two discounts represent the relative distances between cavitation bubbles and solid particles under the condition of applying or not applying the ultrasonic field, respectively. The abscissa represents the observation time, and the event of applying sound field is taken as time 0. After that, each frame interval is 20 μs, and the ordinate represents the distance. According to the picture results, it can be known that, after applying the sound field, the solid particles gradually appear close to the direction of the cavitation bubble under the influence of the micro-jet generated by the collapse of the cavitation bubble, resulting in the relative position between the cavitation bubble and the solid particles shortening to 60% of the original distance. In contrast, in the control group without sound field, the relative distance between cavitation bubbles and solid particles changed little, and the solid particles did not show a trend of moving towards the wall or cavitation bubbles.

## 5. Conclusions

In this paper, the Navier–Stokes equation and the VOF model were used to perform finite element numerical simulations of gas–liquid two-phase motion, and the PISO algorithm was used for the calculations. The optimal parameter values of the cavitation phenomenon were obtained, and ultrasonic cavitation of the bubble near the wall was observed. The experimental platform proves that the cavitation effect affects the trajectory of the solid particles in the microchannel:(1)To address the problem of fluid mixing during ultrasonic cavitation, based on ultrasonic cavitation theory, a new study on the motion trajectory of fine particles after the generation of high-speed directional microjets by ultrasonic cavitation collapse was proposed and simulated by software. The experimental verification proved the feasibility of the research.(2)Through the analysis of the numerical simulation results, the best parameter values for the experiment were obtained, including the initial radius of the cavitation bubble, the ultrasonic frequency, the ultrasonic pressure, and dimensionless quantity, as well as the optimal parameters for the sound field. When the initial bubble radius *R* = 100 μm, *γ* = 1.6, the sound pressure intensity is 300 kPa, and the ultrasonic frequency *f* = 20 kHz, the maximum microjet velocity is generated. Therefore, the working condition under this parameter combination is selected as an example to analyze the characteristics of the bubble burst process, and experimental observation is carried out.(3)An ultrasonic cavitation observation test was used to verify the collapse process of a single bubble. At the same time, the superabsorbent polymerization particles were placed, and fixed hooks were used to facilitate observation. Microfluidic emission caused by bubble cavitation led to deformation of particles, which provided a realistic basis for the viewpoint that cavitation in the flow passage affected the movement trajectory of solid particles.(4)Through the comparative test of whether or not to apply the sound field, the upper part of the cavitation bubble wall sinks downward and collapses, forming a high-speed micro-jet and driving the particles near the top to move quickly to the bottom. The test results are consistent with the theoretical analysis.

Based on a comparison of the results obtained with and without the sound field applied, the upper part of the bubble wall of the cavitation bubble collapsed after the sound field was applied, forming a high-speed microjet and driving the particles near the upper part of the bubble to move quickly towards the bottom surface. The test results and the theory are consistent.

## Figures and Tables

**Figure 1 micromachines-13-01196-f001:**
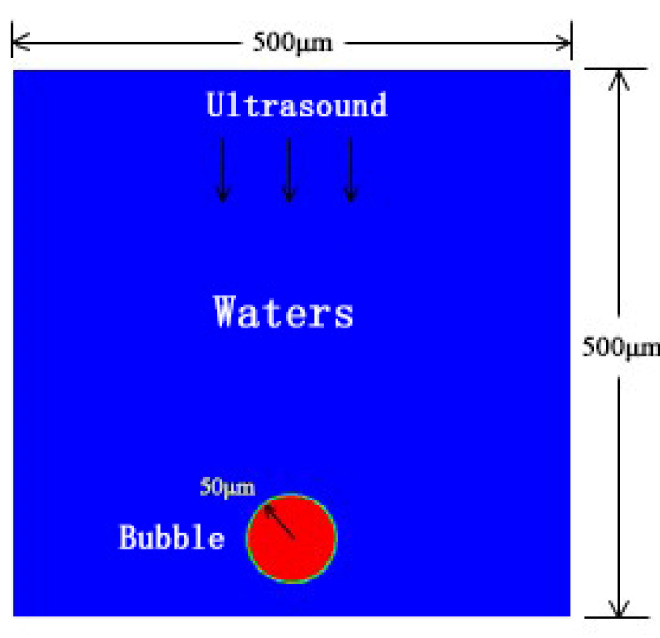
Simulation physical model.

**Figure 2 micromachines-13-01196-f002:**
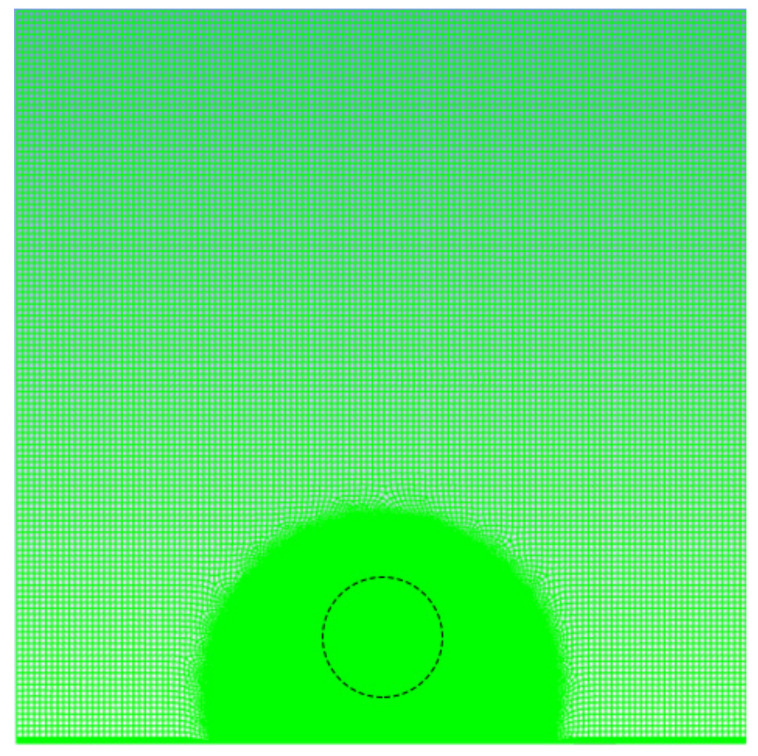
Mesh division.

**Figure 3 micromachines-13-01196-f003:**
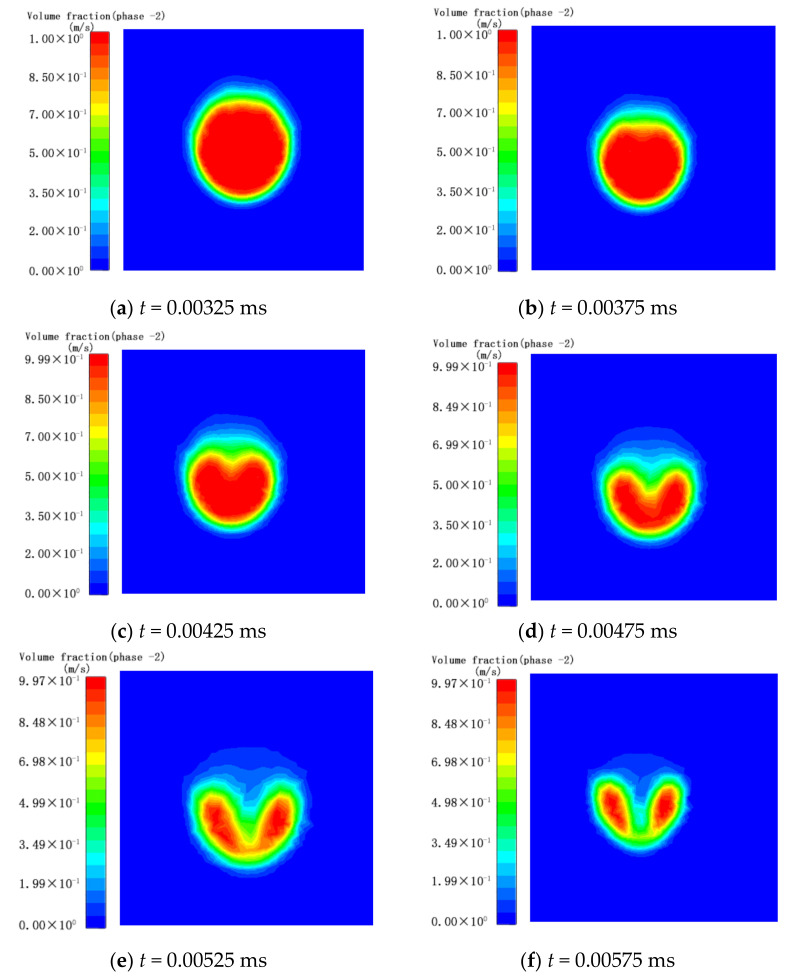
Simulation results of ultrasonic cavitation of bubbles near the wall.

**Figure 4 micromachines-13-01196-f004:**
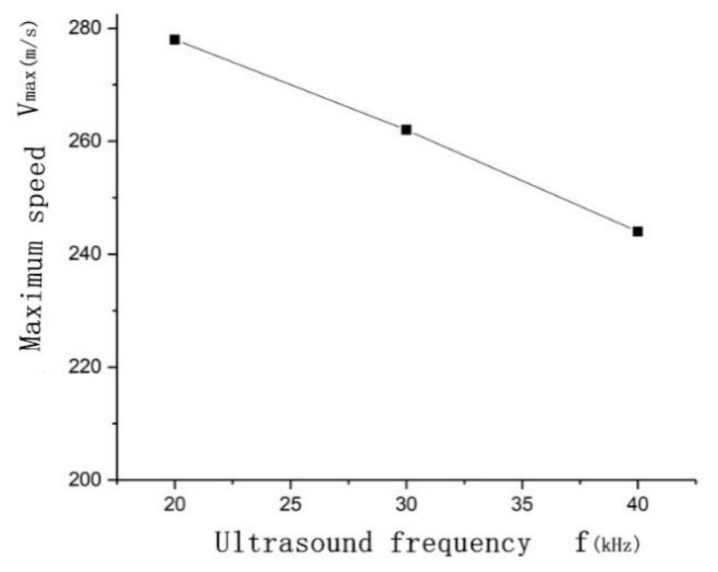
The effect of the initial radius of the bubble on the maximum velocity of the microjet.

**Figure 5 micromachines-13-01196-f005:**
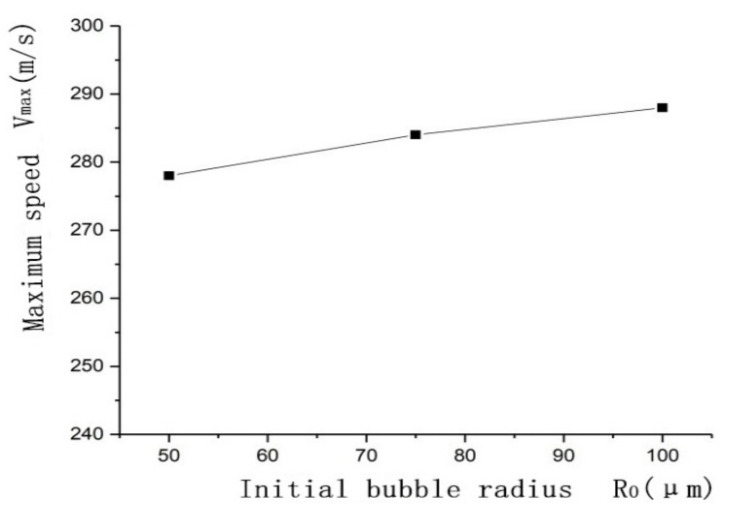
The effect of the ultrasonic frequency on the maximum velocity of the cavitation bubble microjet.

**Figure 6 micromachines-13-01196-f006:**
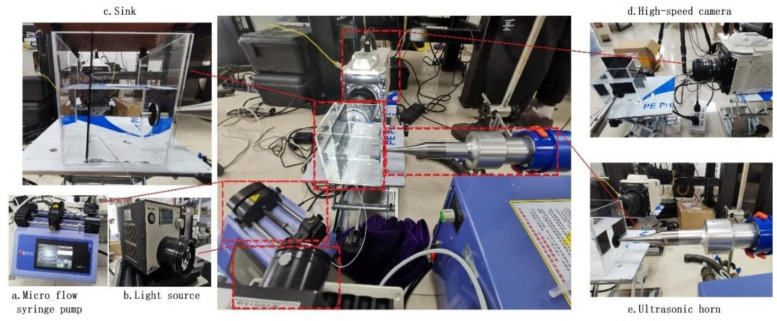
Experimental platform for ultrasonic cavitation observation of bubbles near the wall. (**a**) microfluidic syringe pump; (**b**) light source; (**c**) water tank; (**d**) high-speed camera; (**e**) ultrasonic horn.

**Figure 7 micromachines-13-01196-f007:**
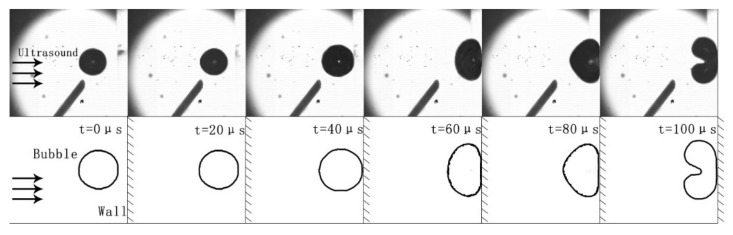
Ultrasonic cavitation collapse process of a single bubble.

**Figure 8 micromachines-13-01196-f008:**
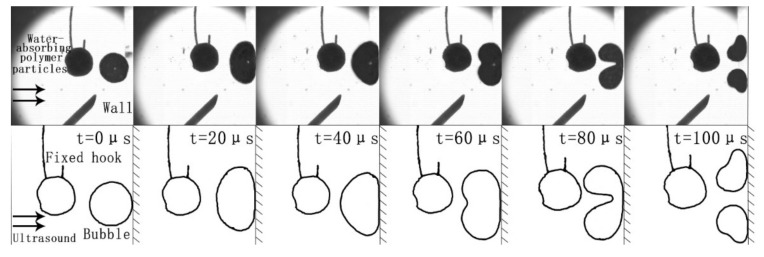
The impact of a single bubble collapse on surrounding particles.

**Figure 9 micromachines-13-01196-f009:**
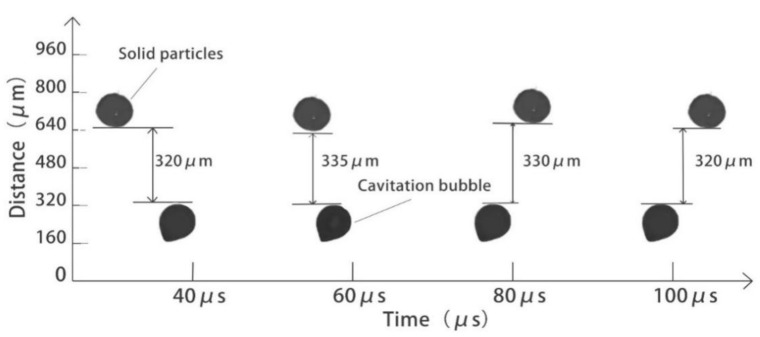
The relative distance between particle and cavitation bubble in flow passage after ultrasonic application.

**Figure 10 micromachines-13-01196-f010:**
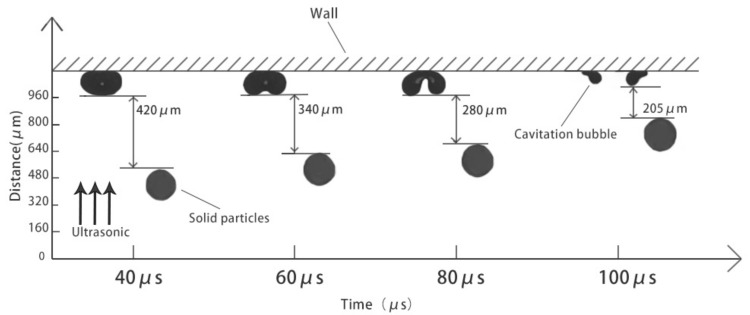
The relative distance between the bubbles and particles in the microfluidic channel when ultrasonic waves are applied.

**Figure 11 micromachines-13-01196-f011:**
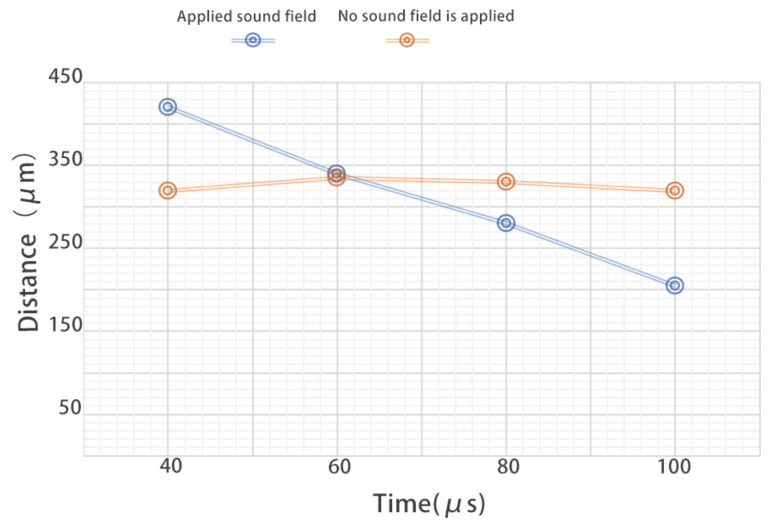
Comparison of the relative distance between the cavitation bubbles and the solid particles under different conditions.

**Table 1 micromachines-13-01196-t001:** Simulation conditions of ultrasonic cavitation near the wall.

Serial Number	Initial Bubble Radius *R*_0_ (μm)	Ultrasound Frequency *f* (kHz)
1	50	20
2	50	30
3	50	40
4	75	20
5	100	20

## Data Availability

The datasets used or analyzed during the current study are available from the corresponding author on reasonable request.

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
