# Peer review of "Study on the Motion Characteristics of Solid Particles in Fine Flow Channels by Ultrasonic Cavitation"

_micromachines, 2022, doi:10.3390/mi13081196_

Round 1

Reviewer 1 Report

Study on the motion characteristics of solid particles in fine flow channels by ultrasonic cavitation is very interesting paper. Some improvement are required

Introduction:

A comparison of particle motion with and without ultrasonic waves proved the feasibility of using the ultrasonic cavitation effect to guide small particles towards the wall.

1.       What is particle size of small particles?

2.       What is size of bubbles?

3.       What is a frequency of an ultrasound field?

Keywords: Fine particles (nanosized or submicron particles?)

What is with an application of ultrasonic cavitation in synthesis of nanosized particles

2. Numerical Theoretical Analysis of Cavitation Bubbles

2.1. Cavitation bubble

Thefluid turbulent (The fluid turbulent)

Please to write a meaning for each symbol in the Equation 1:

2.2.2. Effect of the cavitation bubble radius on cavitation microfluidic jets

When the bubble radius is 5 μm or 500 μm, the bubble wall will periodically oscillate under the action of ultrasound will not collapse.

How are obtained these values? Can be controlled this bubble radius via ultrasonic frequency?

What is morphology of a Bubble (Spherical , cylindrical,…)

2.2.3. The effect of an ultrasonic sound field on cavitation microfluidic jets

The higher the ultrasonic frequency is, the more difficult it is for cavitation to occur!  This is not correct! Please to include these papers in your references:

Bogovic, J, Schwinger, A., Stopic, S., Schroeder, J., Gaukel, V., Schuhmann, P., Friedrich, B. (2011): Controlled droplet size distribution in ultrasonic spray pyrolysis, Metall, 10, 455-459.

Stopic, S., Wenz, F., Volkov-Husovic, T., Friedrich, B. Synthesis of Silica Particles Using Ultrasonic Spray Pyrolysis Method, Metals 2021, 11, 463. https://doi.org/10.3390/met11030463

The cavitation nucleus cannot expand to a sufficient size (what is minimal sufficient size?)

Common frequencies are between 20 and 40 kHz (what is an expected radius of bubble in this range?)

3.1.  Model building

Are the values in the Table 1 based at literature and experimental values? Please to add it in your text.

3.2.  Platform Basic meshing and calculation setting

The dotted line indicates the location of the bubble, and the area near the bubble. (

Figure 2. Mesh division (Please to give an information about Radius of bubble at the Figure 2

Text for „Figure 4. The effect of the ultrasonic frequency on the maximum velocity of the cavitation bubble microjet“ belongs to Fig. 5.

Text for „Figure 5. The effect of the initial radius of the bubble on the maximum velocity of the microjet“ belongs to Fig. 4.

Please to change it.

Page 11: and the solid particle (which type pf solid particles? Metallic Particles?)

Conclusion

The test results and the theory are consistent. Please to give real values for the consistent results (frequency, bubble radius, solid particle size?)?

Author Response

Dear professor:

First of all, thanks for your suggestions for the paper “Study on the motion characteristics of solid particles in fine flow channels by ultrasonic cavitation”, the paper has been revised based on your comments.The revisions to this article are shown in the attached document.

Reviewer 2 Report

After reviewing your article, the overall performance is interesting.

The grammar and content must be revised more. For the first abbreviation, the full name list must be shown, like VOF in page 2. Its full name is shown in page 3. It’s not correct. In page 3, the reference 3 shows the wrong arrangement. It should be “[3].”, not “.[3]”. Couples of grammar issues must be clarified more. Some of sentences are Chinese styles, not English description. The captions in figures are too blurry to be seen. In Fig. 3, the description sentences in page 7 is not suitable.

The overall writing quality must be revised, especially in part 4. By the way, the caption description in Fig. 10 is not suitable. We didn’t see the results and discussion part.

In the technical part, I have some concerns and you must step by step illustrate them.

1. In equations, the meanings of variables must be labeled more. The readers hardly read and sense them.

2. You should add one reference to demonstrate the Euler’s method.

3. In Fig. 4, the error bar must be added to know the deviation in experiment.

4. In Figs. 7 and 8, the max. frequency is 40 kHz, but the experimental data show 20 us, which means the frequency is 50 kHz. Please change it to 40kHz as 25 us. It will be consistent with your content.

5. The difference between simulation and experimental data must be illustrated by some tables or figures. The authors claim that their performance is hugely improved, but I didn’t see that. Please use some concise ways to expose them.

6. In Fig. 11, the trend distributions between both are different. Please comment the mechanism or effect. What is the main contribution with the applied sound field?

7. In Conclusions, the authors mentioned “Through the analysis… for experimental verification”. Please fully illustrate them more in the relationship. Based on the provided consequences, the conclusion is not richly expressed.

Author Response

(The authors gave the same response as above.)

Round 2

Reviewer 2 Report

The revised article can be acceptable, but the English quality must be improved.